# Monitoring ATP dynamics in electrically active white matter tracts

Andrea Trevisol[1], Aiman S Saab[1,2,3], Ulrike Winkler[4], Grit Marx[4], Hiromi Imamura[5], Wiebke Möbius[1,6], Kathrin Kusch[1], Klaus-Armin Nave[1]*, Johannes Hirrlinger[1,4]*

[1]Department of Neurogenetics, Max-Planck-Institute for Experimental Medicine, Göttingen, Germany; [2]Institute of Pharmacology & Toxicology, University of Zurich, Zurich, Switzerland; [3]Neuroscience Center Zurich, University of Zurich, Zurich, Switzerland; [4]Carl-Ludwig-Institute for Physiology, University of Leipzig, Leipzig, Germany; [5]Graduate School of Biostudies, Kyoto University, Kyoto, Japan; [6]Center Nanoscale Microscopy and Molecular Physiology of the Brain, Göttingen, Germany

**Abstract** In several neurodegenerative diseases and myelin disorders, the degeneration profiles of myelinated axons are compatible with underlying energy deficits. However, it is presently impossible to measure selectively axonal ATP levels in the electrically active nervous system. We combined transgenic expression of an ATP-sensor in neurons of mice with confocal FRET imaging and electrophysiological recordings of acutely isolated optic nerves. This allowed us to monitor dynamic changes and activity-dependent axonal ATP homeostasis at the cellular level and in real time. We find that changes in ATP levels correlate well with compound action potentials. However, this correlation is disrupted when metabolism of lactate is inhibited, suggesting that axonal glycolysis products are not sufficient to maintain mitochondrial energy metabolism of electrically active axons. The combined monitoring of cellular ATP and electrical activity is a novel tool to study neuronal and glial energy metabolism in normal physiology and in models of neurodegenerative disorders.

*For correspondence: nave@em.mpg.de (K-AN); johannes.hirrlinger@medizin.uni-leipzig.de (JH)

## Introduction

In the vertebrate nervous system, long axons have emerged as the 'bottle neck' of neuronal integrity (*Coleman and Freeman, 2010*; *Nave, 2010*; *Hill et al., 2016*). In neurodegenerative diseases, axons often reveal first signs of perturbation, e.g. spheroids, long before the pathological hallmarks of disease are seen in neuronal somata, such as large protein aggregates or neuronal cell death. In white matter tracts, Wallerian degeneration of axons is a carefully regulated program of self-destruction that upon reaching a threshold leads to rapid calcium-dependent proteolysis of axonal proteins. This degeneration can be experimentally triggered in different ways, including acute axonal dissection or by blocking mitochondrial energy metabolism (*Coleman and Freeman, 2010*; *Hill et al., 2016*). In disease situations, the mechanisms leading to axon loss are much less clear. For example, in spastic paraplegia (SPG), which comprises an entire family of inherited axon degeneration disorders, long myelinated tracts of the spinal cord are progressively lost. However, virtually none of the subforms shows a clear link between the primary defect and the later loss of myelinated axons, with the possible exception of mitochondrial disorders, in which the perturbation of axonal energy metabolism is assumed to underlie a loss of axonal ATP (*Ferreirinha et al., 2004*; *Tarrade et al., 2006*). However, a disturbed energy balance has never been shown. Similarly, in the equally heterogeneous group of Charcot-Marie-Tooth (CMT) neuropathies of the peripheral nervous system only a small subset is caused by mitochondrial dysfunction. In most patients (CMT1A) the primary defect resides in the

**eLife digest** The brain contains an intricate network of nerve cells that receive, process, send and store information. This information travels as electrical impulses along a long, thin part of each nerve cell known as the nerve fiber or axon. The act of sending these electrical signals requires a lot of energy, and energy in cells is most often stored within molecules of adenosine triphosphate (called ATP for short).

Importantly, a better understanding of how the production and consumption of ATP in nerve cells relates to electrical activity would help scientists to better understand how a shortage of energy in the brain contributes to diseases like multiple sclerosis. However, to date, it has been challenging to study the dynamics of ATP in nerve cells that are active.

Now, Trevisiol et al. describe a new system that allows changes in ATP levels to be seen within active nerve cells. First, mice were genetically engineered to produce a molecule that works like an ATP sensor only in their nerve cells. This made it possible to visualize the amount of ATP inside the axons in real-time using a microscope. Measuring ATP levels and recording the electrical signals moving along an axon at the same time allowed Trevisiol et al. to see how ATP content and electrical activity correlate and regulate each other.

The experiments reveal that strong electrical activity reduces the ATP content of the axon. Trevisiol et al. also discovered that nerve cells are unable to generate enough energy on their own to sustain their electrical activity. These results provide evidence that other cells in the brain – most likely non-nerve cells called oligodendrocytes – play an active role in delivering energy-rich substances to the axons of nerve cells.

In the future, the same tools and approaches could be used to monitor ATP levels and electrical activity in mice that model neurological disorders. Such experiments could tell scientists more about how disturbing energy production in nerve cells affects these diseases.

axon-associated myelinating glia (Schwann cells), and it is unknown whether the axonal energy metabolism is perturbed before the onset of axon degeneration.

Axonal ATP consumption in white matter tracts largely depends on the electric spiking activity that differs between regions, and it is virtually impossible to biochemically determine ATP selectively in the axonal compartment and under 'working conditions' with a high temporal and spatial resolution. To solve this problem, we have developed a novel approach to simultaneously study action potential propagation and ATP levels in axons of optic nerves of mice, a white matter tract suitable for studying axonal energy metabolism (*Stys et al., 1991*; *Saab et al., 2016*). We established the transgenic mouse line ThyAT, in which the fluorescent ATP-sensor ATeam1.03$^{YEMK}$ (*Imamura et al., 2009*) shows pan-neuronal expression in vivo. Stimulating electrically optic nerves acutely isolated from those mice, we combined recordings of compound action potentials (CAP) with confocal imaging of the ATP-sensor to evaluate and compare both axonal conductivity and ATP levels. We find that (1) axonal glycolysis is not sufficient to robustly sustain CAPs and physiological ATP levels, but mitochondrial function is needed to provide ATP; (2) during high-frequency propagation of action potentials, CAPs are a well-suited parameter to estimate axonal ATP levels; (3) lactate metabolism is essential for the maintenance of axonal ATP levels, and its inhibition severely affects energy homeostasis of myelinated axons.

## Results

### ATP imaging in optic nerves

The genetically encoded ATP-sensor ATeam1.03$^{YEMK}$ allows FRET-based monitoring of cellular ATP levels close to real-time (*Imamura et al., 2009*). We generated transgenic mice for pan-neuronal in vivo expression of this sensor driven by the murine *Thy1.2* promoter (*Caroni, 1997*). One mouse line with 21 copies of the transgene [B6-Tg(Thy1.2-ATeam1.03$^{YEMK}$)AJhi, referred to as ThyAT] displayed widespread neuronal expression (*Figure 1*). In the retina, fluorescence marked ganglion cells and

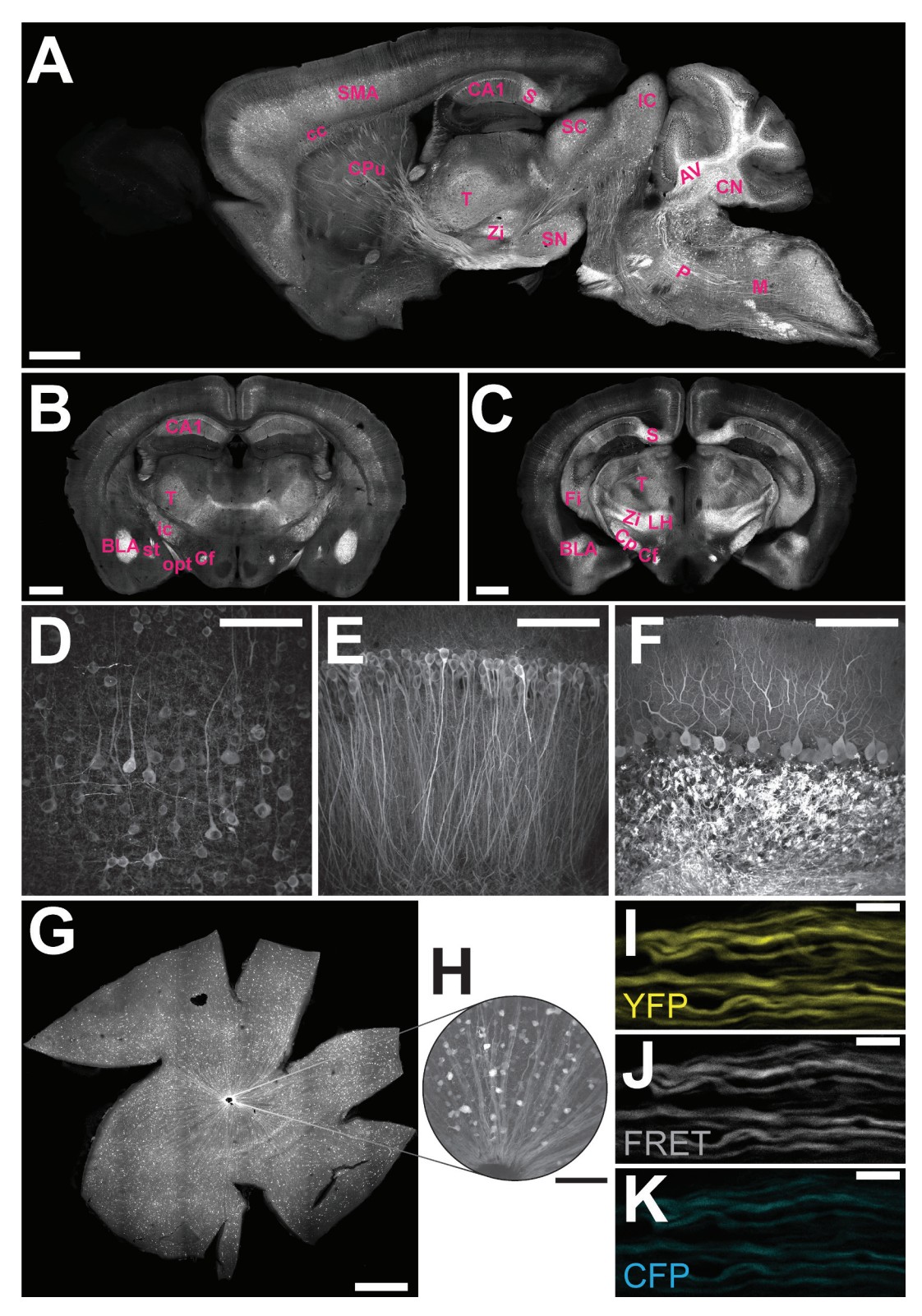

**Figure 1.** Characterization of the expression pattern of the newly generated B6-Tg(Thy1.2-ATeam1.03^YEMK)AJhi (ThyAT)-mouse line. (**A**) Sagittal section of the brain highlights broad ATeam1.03^YEMK expression in neurons in almost all brain regions with the exception of the olfactory bulb. Scale bar: 1 mm. (**B,C**) ThyAT expression pattern in coronal brain sections revealing sensor expression e.g. in thalamus, hypothalamus, amygdala, cortex and hippocampus. Scale bar: 1 mm. Abbreviations used in panels **A–C** are: AV: arbor vitae; BLA: basolateral amygdalar nucleus, anterior; CA1: CA1 region

*Figure 1 continued on next page*

Figure 1 continued

of the hippocampus; cc: corpus callosum; Cf: columns of the fornix; CN: cerebellar nuclei; Cp: cerebral peduncle; CPu: caudate putamen; Fi: fimbria; IC: inferior colliculus; ic: internal capsule; LH: lateral area of the hypothalamus; M: medulla; opt: optic tract; P: pons; S: subiculum; SC: superior colliculus; SMA: somato-motor area (cortex); SN: substantia nigra; st: stria terminalis; T: thalamus; Zi: zona incerta (thalamus). (D) Within the cortex, neurons expressing ATeam1.03$^{YEMK}$ are clearly visible including their processes. Note the lack of ATP-sensor localization to the nucleus. Scale bar: 100 µm. (E) Also in the hippocampus neurons strongly express ATeam1.03$^{YEMK}$. Scale bar: 100 µm. (F) In the cerebellum, Purkinje cells express the ATP-sensor. In addition, incoming mossy fibers strongly express ATeam1.03$^{YEMK}$. Scale bar: 100 µm. Images in panels A, D–F are obtained on brain slices from a four month old animal, images in panels B and C are from mice at the age of two month. (G) Expression pattern of the ATP-sensor in the retina. *Thy1.2* promoter drives the expression of ATeam1.03$^{YEMK}$ in ganglion cells. Scale bar: 1 mm. (H) Magnified view of neurons and axons in the retina expressing ATeam1.03$^{YEMK}$. Scale bar: 100 µm. (I–K) Representative images of optic nerve axons showing the YFP channel (I), FRET channel (J) and CFP channel (K). The ATeam1.03$^{YEMK}$ expression is present in different axons independent of their diameter. Scale bar: 10 µm.

their axons (*Figure 1G,H*) and expression of the ATP-sensor in myelinated axons appeared robust in all recorded channels (*Figure 1I–K*).

Optic nerves from adult ThyAT mice were studied ex vivo, assessing axonal ATP levels by confocal microscopy and simultaneously monitoring stimulus-evoked CAPs (*Figure 2*). To verify sensor function, nerves were subjected to ATP depletion by blocking mitochondrial respiration with sodium azide (MB) and glucose deprivation (GD). An immediate drop of the FRET signal (*Figure 2C,D*) and increase in CFP emission (*Figure 2C,D*) indicated that ATP levels in axons dropped. Changes in pH can modulate YFP-fluorescence (*Nagai et al., 2004*; *Zhao et al., 2011*). However, ATeam1.03$^{YEMK}$ is almost insensitive to pH within the physiological range (*Imamura et al., 2009*; *Surin et al., 2014*) and YFP emission upon direct YFP excitation was unchanged (*Figure 2C,D*; and data not shown), suggesting that pH changes are not the cause of altered FRET signals.

The ratio of FRET/CFP fluorescence (F/C-ratio) was calculated as a measure for the cytosolic ATP concentration (*Imamura et al., 2009*; *Figure 2E,F*). The baseline F/C-ratio (control condition; 10 mM glucose) was similar and stable between different nerves analyzed (*Figure 2F*), while after 2 min of MB+GD the F/C-ratio was reduced by $35.1 \pm 1.7\%$ (p<0.001; n = 19 nerves; *Figure 2F*). In further experiments, F/C-ratios were normalized between one (10 mM glucose) and zero (MB+GD; *Figure 2G*). As expected, during MB+GD the decrease in axonal ATP was mirrored by a decay of CAPs, reflecting conduction blocks (*Figure 2H*). Strikingly, also the decline rates of axonal ATP and conductivity were similar (*Figure 2I*).

## Kinetics of ATP decline during energy deprivation

Next, we compared ATP and CAP dynamics in response to different modes of energy deprivation (*Figure 3*; *Figure 3—figure supplement 1*), including glucose deprivation (GD), blockage of mitochondrial respiration by azide (MB), and a combination of both (MB+GD). During GD both ATP and CAPs decayed after 13 min (*Figure 3A*). Inhibition of respiration resulted in a drop after about 3 min with a steeper decline (*Figure 3B*). MB+GD induced an immediate and fast decline of ATP, followed by an equally fast decline in CAPs (*Figure 3C–E*). To reduce permanent damage and to study recovery, MB and MB+GD were limited to 5 min. The observed change of the ATP sensor signal is most likely not caused by major changes in pH as YFP fluorescence remained almost unchanged under all conditions of energy deprivation (*Figure 2D* and *Figure 3—figure supplement 2*). These data suggest that optic nerves continue ATP production under GD, presumably by metabolizing glycogen of astrocytes (*Brown et al., 2005*). However, axonal ATP production strongly depends on mitochondrial respiration, and axonal glycolysis alone is insufficient to maintain ATP levels, even in the virtual absence of electrical activity (baseline recording conditions with 0.033 Hz stimulation frequency).

When nerves were reperfused (aCSF, 10 mM glucose) after GD, CAPs recovered with a delay of about 3 min, and the recovery of CAPs preceded the visible recovery of ATP levels by several minutes (*Figure 3A,F*). Interestingly, following MB+GD, CAPs recovered significantly sooner, concomitantly with axonal ATP, and faster (*Figure 3F,G*). We presume that, under MB+GD, lactate/pyruvate was not depleted from the optic nerve and is readily available for ATP generation by mitochondria after cessation of MB+GD.

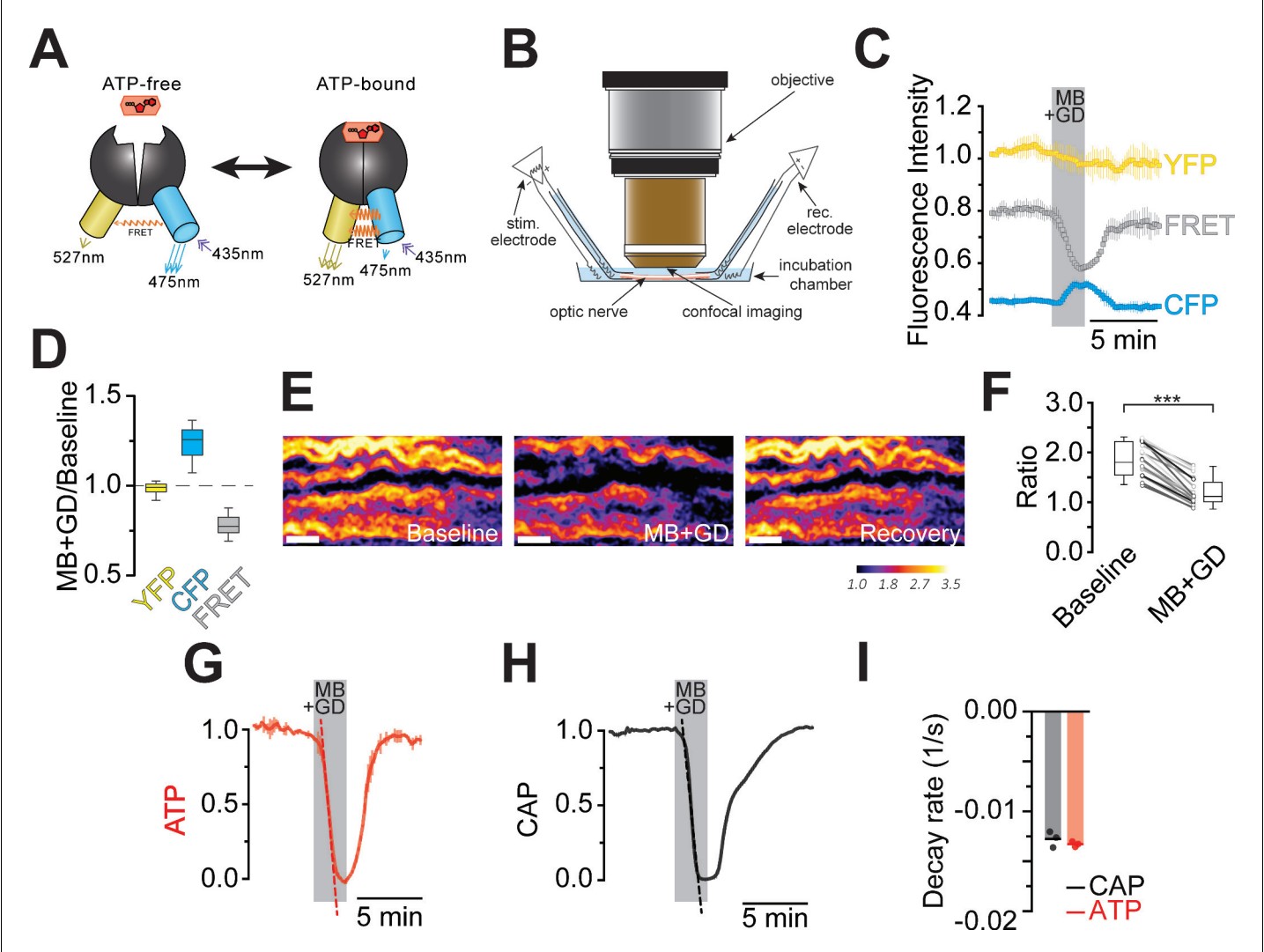

**Figure 2.** Imaging of ATP combined with electrophysiology in acutely isolated optic nerves of ThyAT-mice. (A) Binding of ATP induces a conformational change in the genetically encoded ATP-sensor ATeam1.03[YEMK] thus increasing the FRET effect (YFP emission upon CFP excitation) and simultaneous decreased emission of CFP (upon CFP excitation). The ratio between FRET and CFP can thus be correlated with the concentration of ATP present in the cell. (B) Schematic representation of the set-up to acquire evoked CAPs in the optic nerve and to simultaneously investigate relative ATP levels by electrophysiology and confocal imaging, respectively. (C) Time course of fluorescence intensity recorded in the YFP, FRET and CFP- channels during application of mitochondrial blockage (MB) and glucose deprivation (GD) for 2.5 min. Values are normalized to YFP intensity prior to application of MB+GD (n = 3 nerves). Time resolution: 10.4 s. (D) The combination of MB and GD is a fast and reliable way to deplete ATP in axons of the optic nerve. ATP depletion is measured as a decrease in FRET and increase in CFP, calculated as ratio between fluorophore intensity during MB+GD, over fluorophore intensity at baseline condition (MB+GD/Baseline). Notably, YFP emission upon YFP excitation remains unchanged (n = 5 nerves). (E) Ratiometric images displaying the FRET/CFP ratio of the ATeam1.03[YEMK]–sensor in the axons of the optic nerve during ATP depletion following MB+GD. The phases before (Baseline) and after (Recovery) are also shown. Scale bar: 10 μm. (F) FRET/CFP ratio values (not normalized) during baseline and MB+GD. The boxplots show summarized data of n = 19 nerves, lines in between boxplots show changes in the FRET/CFP ratio of all 19 individual nerves (***p<0.001). (G) To assess ATP variations, the ratio of the fluorescence intensities of the FRET and CFP-channel was calculated (FRET/CFP ratio) and normalized to baseline (set as 1) and MB+GD (set as 0). The red dashed line visualizes the slope of ATP drop at the point of maximal velocity of ATP decay during mitochondrial blockage (MB+GD, n = 3). (H) Recording of the evoked compound action potential (CAP), given as the normalized curve integral during mitochondrial blockage and glucose deprivation (MB+GD). The black dashed line represents the slope at the point of maximal velocity of CAP changes during MB+GD treatment (n = 3). Individual CAP traces are shown in *Figure 3—figure supplement 1*. (I) Stripe plot describing CAP (black) and ATP (red) kinetics, expressed as maximal variation per s, during MB+GD (p=0.39, n = 3, Welch's t-test). Dots show individual data points, bars and lines represent the mean of all data.

The following source data is available for figure 2:

*Figure 2 continued on next page*

*Figure 2 continued*

**Source data 1.** Table containing data for *Figure 2*.

## Electrical activity during high frequency stimulation depletes ATP

Maintaining axonal conductivity ex vivo depends on energy-rich substrates and high- frequency stimulation of axons causes CAPs to decrease (*Brown et al., 2001*; *Tekkök et al., 2005*). However, it has never been possible to study the reverse, i.e. the impact of electrical activity and spiking frequency on axonal ATP levels. In ThyAT optic nerves incubated in aCSF containing 10 mM glucose and subjected to electrical stimulation at 16 Hz, 50 Hz or 100 Hz for 2.5 min, CAPs dropped in a stimulation frequency dependent manner (16 Hz: 94.9 ± 1.5%; 50 Hz: 79.6 ± 0.4%; 100 Hz: 66.4 ± 1.0%; n = 5 nerves; *Figure 4A,C*, *Figure 4—figure supplement 1*). At the same time, axonal ATP levels decreased as stimulation frequencies increased (16 Hz: 91.6 ± 1.0%; 50 Hz: 82.3 ± 1.4%; 100 Hz: 68.9 ± 2.8%; n = 5 nerves; *Figure 4B,D*) indicating higher axonal ATP consumption. Even maximal ATP changes and maximal loss of conductivity correlated over stimulation frequencies (*Figure 4E,F*). Similar results were obtained with a different stimulation paradigm with continuously increasing frequencies (*Figure 4—figure supplement 2*); however, the causal relations of the correlation of CAP and ATP changes are most likely more complex (see discussion below). Finally, both the rates of ATP drop and recovery correlated with the amplitude of ATP changes (*Figure 4—figure supplement 3*).

## ATP homeostasis of myelinated axons depend on lactate metabolism

Next, we asked whether lower (physiological) glucose concentrations have an impact on nerve conduction and axonal ATP levels when nerves were challenged with different spiking frequencies (*Figure 4C,D*; *Figure 4—figure supplement 4*). In presence of 10 mM and 3.3 mM glucose, CAP performance and ATP levels dropped similarly with increasing stimulation frequencies. However, when only 2 mM glucose was applied both CAP and ATP levels were substantially decreased (*Figure 4C,D*) and the ATP/CAP ratio was more variable at 2 mM glucose (*Figure 4F*). Analysis of the fluorescence intensity of single channels indicated that the observed changes of the ATP sensor signal are indeed caused by ATP changes because YFP fluorescence remained rather unaffected (*Figure 4—figure supplement 5*). Taken together, these data suggest that 2 mM glucose is below the threshold required for maintaining the axonal energy balance of the optic nerve in the ex vivo experimental setting.

At high-frequency stimulation and under aerobic conditions, optic nerve CAPs were maintained equally well with either 10 mM glucose, 10 mM lactate or 10 mM pyruvate as extracellular energy substrate (*Figure 5A*), in agreement with earlier findings (*Brown et al., 2001*; *Tekkök et al., 2005*) presumably because absolute axonal ATP consumption is unaffected by the type of metabolic support. However, at 100 Hz the relative decrease in axonal ATP levels was larger with lactate (or pyruvate) than with glucose (glucose: 68.9 ± 2.8%; lactate: 56.3 ± 2.8%; pyruvate: 47.4 ± 2.8%; p=0.05 and p=0.01, respectively; n = 3 nerves; *Figure 5B*, *Figure 5—figure supplement 1*), showing that 10 mM glucose maintains axonal ATP production better than 10 mM lactate or pyruvate if applied exogenously.

Do axons require pyruvate/lactate metabolism also in the presence of glucose as energy substrate? To address this question, we studied optic nerves conduction and ATP levels in 3.3 mM glucose plus 20 mM D-lactate, which competitively inhibits transport through MCTs as well as lactate dehydrogenase (LDH; D-lactate is a stereoisomer of L-lactate, which is the physiological molecule in energy metabolism). D-lactate did not affect CAPs (*Figure 5C*, *Figure 5—figure supplement 2*) but had a strong effect on axonal ATP (at 100 Hz: Glc: 59.7 ± 2.9%; Glc+D-Lac: 29.2 ± 2.2%; p=0.0001; n = 5 nerves; *Figure 5D*). Similarly, when MCT1/MCT2-mediated lactate transport was inhibited using the specific inhibitor AR-C155858 (*Ovens et al., 2010*), relative ATP levels were strongly reduced (at 100 Hz: 24.8 ± 4.1%; p=0.0004; n = 5 nerves), but with no effect on CAP performance (at 100 Hz: Glc: 69.5 ± 4.2%; Glc+AR-C155858: 62.8 ± 3.3%; p=0.37; n = 6 nerves; *Figure 5C,D*; *Figure 5—figure supplement 2*), indicating the presence of a 'safety window'. When lactate metabolism was impaired, ATP/CAP ratios strongly deviated from the normal '1:1 ratio' observed in the

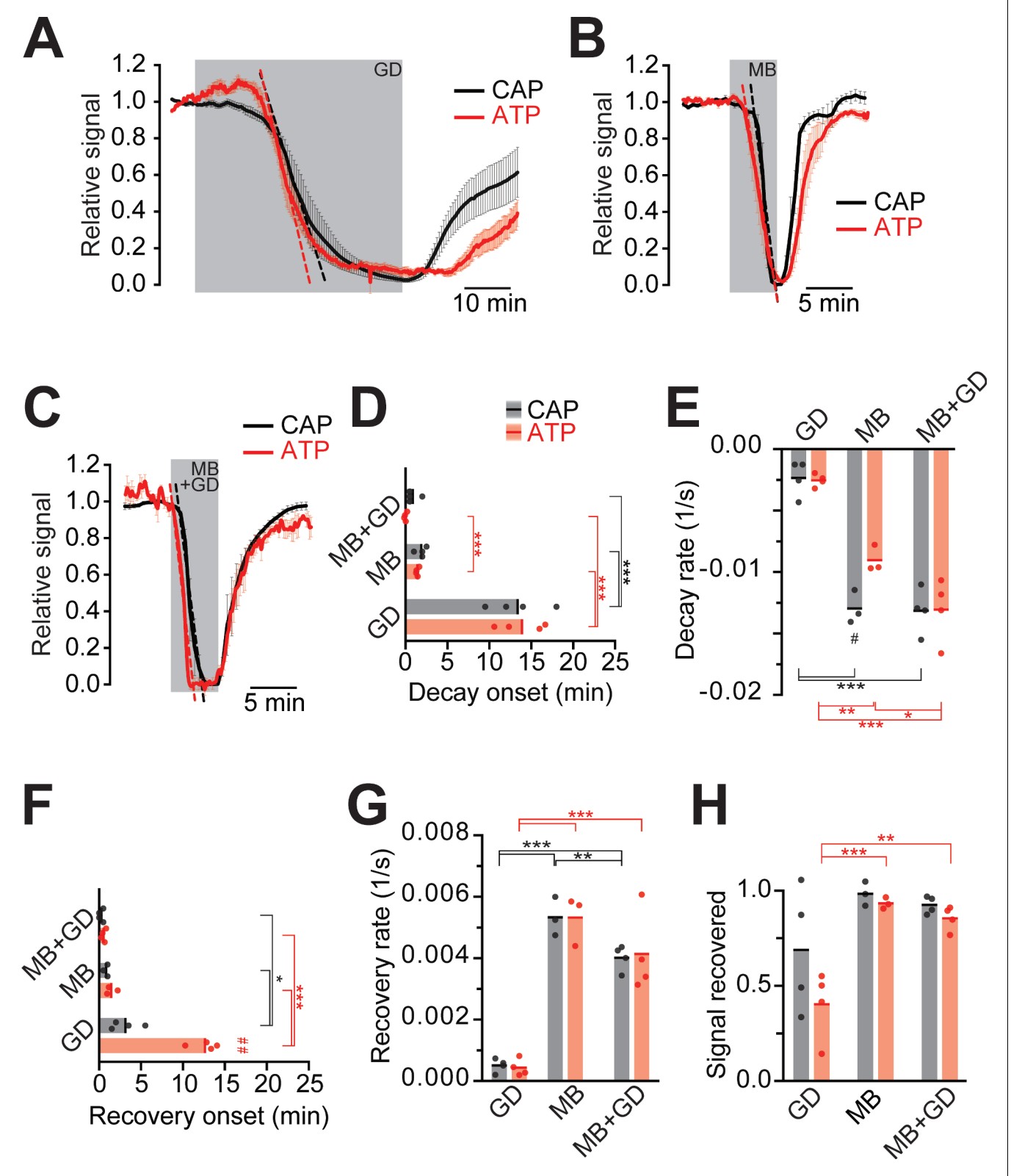

**Figure 3.** Impairment of axonal ATP and CAP by glucose deprivation and/or inhibition of mitochondrial respiration. (**A**) Removal of glucose from the aCSF (glucose deprivation, GD, 45 min) induces similar ATP (red) and CAP (black) decays starting at around 13 min after onset of the treatment. Red and black dashed, straight lines represent the maximum velocity of ATP and CAP decay (also applies to panels **B** and **C**). When 10 mM glucose is restored, CAP recovery precedes ATP restoration (n = 4 nerves). (**B**) Blockade of mitochondrial respiration by azide (MB, 5 min) produces a fast decay in

*Figure 3 continued on next page*

*Figure 3 continued*

ATP (red) and CAP (black) starting at 1.8 min after beginning of treatment. When azide is removed, CAP and ATP are promptly restored, with CAP recovery preceding the ATP increase (n = 4 nerves). (C) Simultaneous removal of glucose and blockade of mitochondrial respiration with azide (MB +GD, 5 min) produces a fast decay in ATP (red) preceding CAP decay (black), starting already 0.5 min after onset of treatment. Following azide removal and replenishment of glucose, CAP and ATP are restored (n = 4 nerves). (D) Time of onset of the ATP or CAP decay. The slowest decay induction was observed during glucose deprivation. (E) Velocity of signal decay for ATP and CAP during each of the three treatments: glucose deprivation (GD), mitochondrial blockage (MB) and the combination of both (MB+GD). (F) Time of onset of ATP or CAP recovery after reperfusion with control aCSF containing 10 mM glucose. (G) Rate of recovery of both ATP and CAP during reperfusion of the nerves with aCSF containing 10 mM glucose after the treatments indicated. (H) Comparison of ATP and CAP area overall recovery after individual treatments. Data in D–H is presented as stripe plots, with dots representing individual data points, bars and lines showing the mean. Hash signs indicate statistically significant differences between ATP and CAP under the same condition (#p<0.05, ##p<0.01, paired t-test); asterisks on red (ATP) and black (CAP) lines indicate statistically significant differences between different conditions (*p<0.05, **p<0.01, ***p<0.001; one-way ANOVA with Newman-Keuls post-hoc test).

The following source data and figure supplements are available for figure 3:

**Source data 1.** Table containing data for *Figure 3*.
**Figure supplement 1.** Example of progression of CAP traces' decay during energy deprivation.
**Figure supplement 2.** Analysis of fluorescence changes of the ATP sensor during application of different models of energy deprivation to optic nerves.

presence of glucose alone (*Figure 5E,F*). Collectively, these results strengthen the conclusion that pyruvate/lactate metabolism is required to maintain ATP levels in fast spiking axons.

## Discussion

The axonal energy balance is determined by the equilibrium of ATP consumption and ATP generation, which is a function of axonal spiking frequency and metabolic support, the latter provided in part by glycolytic oligodendrocytes (*Fünfschilling et al., 2012*; *Lee et al., 2012*; *Saab et al., 2016*). Here, we have generated transgenic mice expressing an ATP-sensor in neurons and in the axonal compartment of myelinated nerves. These mice allowed us to study for the first time the interrelationship of electrical activity and axonal ATP homeostasis in real time by adapting an established optic nerve model of white matter electrophysiology (*Brown et al., 2001*; *Brown and Ransom, 2007*). Specifically, we show that both CAPs and axonal ATP content decrease in a stimulation frequency-dependent manner, and that even in the presence of glucose as (exogenous) energy substrate, lactate contributes to ATP homeostasis. We also analyzed short-term adaptations of metabolism in a model of ischemic stroke when ATP demands are increased, i.e. the recovery phase from transient glucose deprivation and mitochondrial inhibition.

While being a powerful experimental system, some caveats need to be taken into account to interpret the obtained results. First, the signal of the ATeam1.03$^{YEMK}$ sensor is not linearly related to ATP over an extended range of concentrations (*Imamura et al., 2009*). However, in cultured neurons the cytosolic ATP concentration has been estimated at approximately 2 mM (*Rangaraju et al., 2014*; *Pathak et al., 2015*), i.e. within the linear range of ATeam1.03$^{YEMK}$. We are thus confident that the observed changes in sensor signal reflect similar relative changes of axonal ATP. However, as calibration of the ATP-sensor signal to absolute cytosolic concentrations is difficult in the myelinated optic nerve, a precise estimation of the absolute concentration of axonal ATP awaits further technical advances. Nevertheless, changes of the ATP signal in the course of an experiment could be detected even after mild stimulations, which suggest that the basal axonal ATP level is within the dynamic range of the sensor with its $K_D$ of 1.2 mM (*Imamura et al., 2009*).

Secondly, with the spatial resolution of our approach, ATP signals reflect mean cytosolic concentrations in the axoplasm. It has been suggested that local differences of ATP consumption by $Na^+$/$K^+$-ATPase lead to ATP microdomains close to the cell membrane with low ATP concentrations (*Haller et al., 2001*; *Toloe et al., 2014*). Contrarily, ATP diffusion is reported to be very fast not allowing formation of ATP microdomains (*Barros et al., 2013*). While such highly localized changes of ATP would be missed in our experiments, they are unlikely to affect the principle findings.

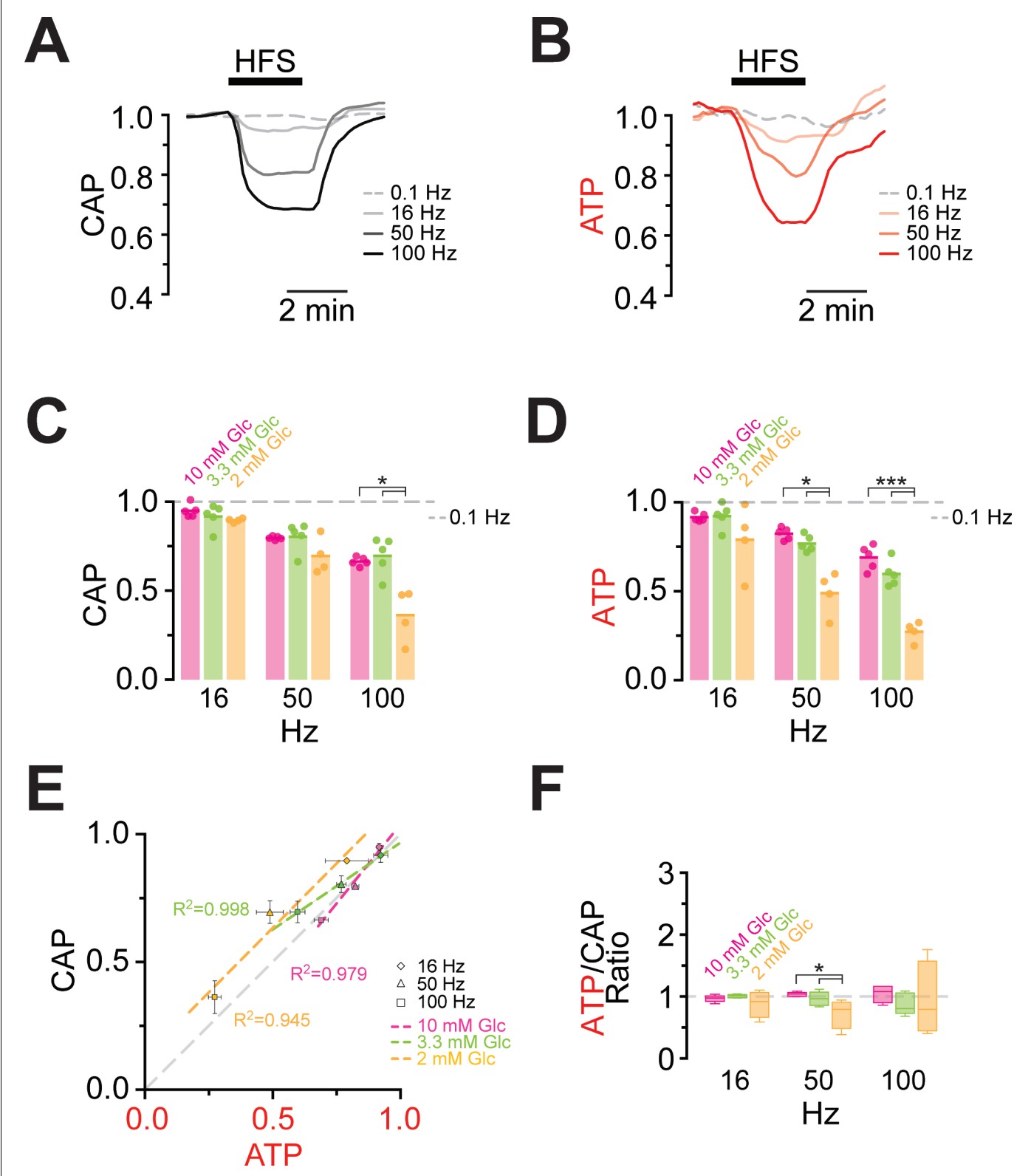

**Figure 4.** Comparison of ATP and CAP dynamics during high frequency stimulation. (**A**) The CAP area decreases over time during high-frequency stimulation (HFS). The decay amplitude deviates from the absence of HFS, indicated by the dashed line (0.1 Hz, used for normalization to 1.0), and increases progressively with the increase in stimulation frequency (16 Hz, 50 Hz, 100 Hz). Traces from one representative nerve incubated in aCSF containing 10 mM glucose are shown. (**B**) Axonal ATP levels also decrease with increasing stimulation frequency, reaching a new steady state level

*Figure 4 continued on next page*

*Figure 4 continued*

which depends on the stimulation frequency. Same experiment as in panel **A**. (**C**) Remaining CAP area at the end of the HFS (overall decay amplitude) during incubation of nerves in different glucose concentrations quantified during the last 30 s of HFS. The stripe plot shows summarized data from n = 5, 5, or 4 nerves for 10 mM, 3.3 mM and 2 mM glucose, respectively. The dashed line at 1 shows CAP size at 0.1 Hz stimulation frequency, which was used for normalization. (**D**) Quantification of ATP decay amplitude during incubation of the same nerves as in (**C**) in different glucose concentrations. The dashed line at 1 shows ATP levels at 0.1 Hz stimulation frequency. (**E**) Correlation of the amplitude of ATP and CAP decay during HFS of nerves bathed in aCSF containing the glucose concentrations indicated. Data points are very close to the diagonal of the graph indicating that ATP and CAP change by similar factors. (**F**) Ratio of ATP and CAP drop during HFS in the presence of glucose in the concentrations indicated. If both parameters change by the same factor, this ratio remains equal to one. Data in (**C–D**) is presented as stripe plots, with dots representing individual data points and bars and lines showing the mean. Asterisks indicate statistically significant differences between glucose concentrations (*p<0.05, ***p<0.001; Welch's t-test).

The following source data and figure supplements are available for figure 4:

**Source data 1.** Table containing data for *Figure 4*.

**Figure supplement 1.** Example of progression of CAP traces' decay during high frequency stimulation (HFS).

**Figure supplement 2.** Stimulation of optic nerves with progressively increasing frequencies.

**Figure supplement 3.** Correlation of the rates and amplitudes of CAP and ATP changes during HFS in different glucose concentrations.

**Figure supplement 4.** Example of CAP traces before and after high-frequency stimulation (HFS) of optic nerves incubated in aCSF with different concentrations of glucose.

**Figure supplement 5.** Analysis of fluorescence changes of the ATP sensor during high-frequency stimulation (HFS) of optic nerves incubated in aCSF with different concentrations of glucose.

Finally, CAPs recorded with supra-threshold stimulation reflect the summation of action potential propagation of virtually all axons within the optic nerve. In contrast, ATP imaging was performed on a single optical plane close to the surface of the nerve comprising typically around 30 axons. Therefore, the observed ATP signals are likely not representative for all axons, as the diffusion of substrates into the core of the nerve may be less efficient than to axons closer to the surface. Our attempts to address this issue by 2-photon laser scanning microscopy deeper within the nerve have failed so far, due to scattering occurring in a densely packed and fully myelinated optic nerve (ASS, unpublished observation). Taking such an inside-out gradient of increasing perfusion efficacy into account, our approach is presently the best possible approximation of the simultaneous monitoring of both parameters. Thus, we are not yet able to assign absolute ATP concentrations to physiological properties of spiking axons. However, we can readily (i) monitor ATP changes as a function of axonal spiking activity, (ii) compare metabolites, (iii) assess the role of disease-causing mutations on axonal conduction and energy metabolism, also under physiological challenge, and finally (iv) study selected drugs and nutrients in axonal energy metabolism to explore treatment strategies.

## Monitoring axonal ATP homeostasis

Targeting the energy producing biochemical pathways by the withdrawal of glucose (GD), blocking mitochondrial respiration (MB), or both (MB+GD), rapidly reduced axonal ATP content, as expected. In GD, ATP fell after 13 min, consistent with the utilization of astroglial glycogen as a reserve fuel (*Brown and Ransom, 2007*). In contrast, MB caused an immediate drop in ATP. In all conditions, ATP decreases preceded or coincided with CAP decay, confirming that energy depletion causes axonal failure (but see caveat above). Interestingly, after GD in the subsequent recovery phase with glucose, the restoration of ATP levels appeared delayed compared to the recovery of conduction. We therefore suggest that at the beginning of reperfusion most newly generated ATP is consumed by Na$^+$/K$^+$-ATPases for reestablishing ion gradients and conduction before it accumulates in the axon and binds to the ATP-sensor.

Astrocytes can mobilize glycogen and generate lactate to support axonal energy homeostasis (*Brown et al., 2001*, *2003*, *2005*). However, it has been a long-standing debate whether neuronal

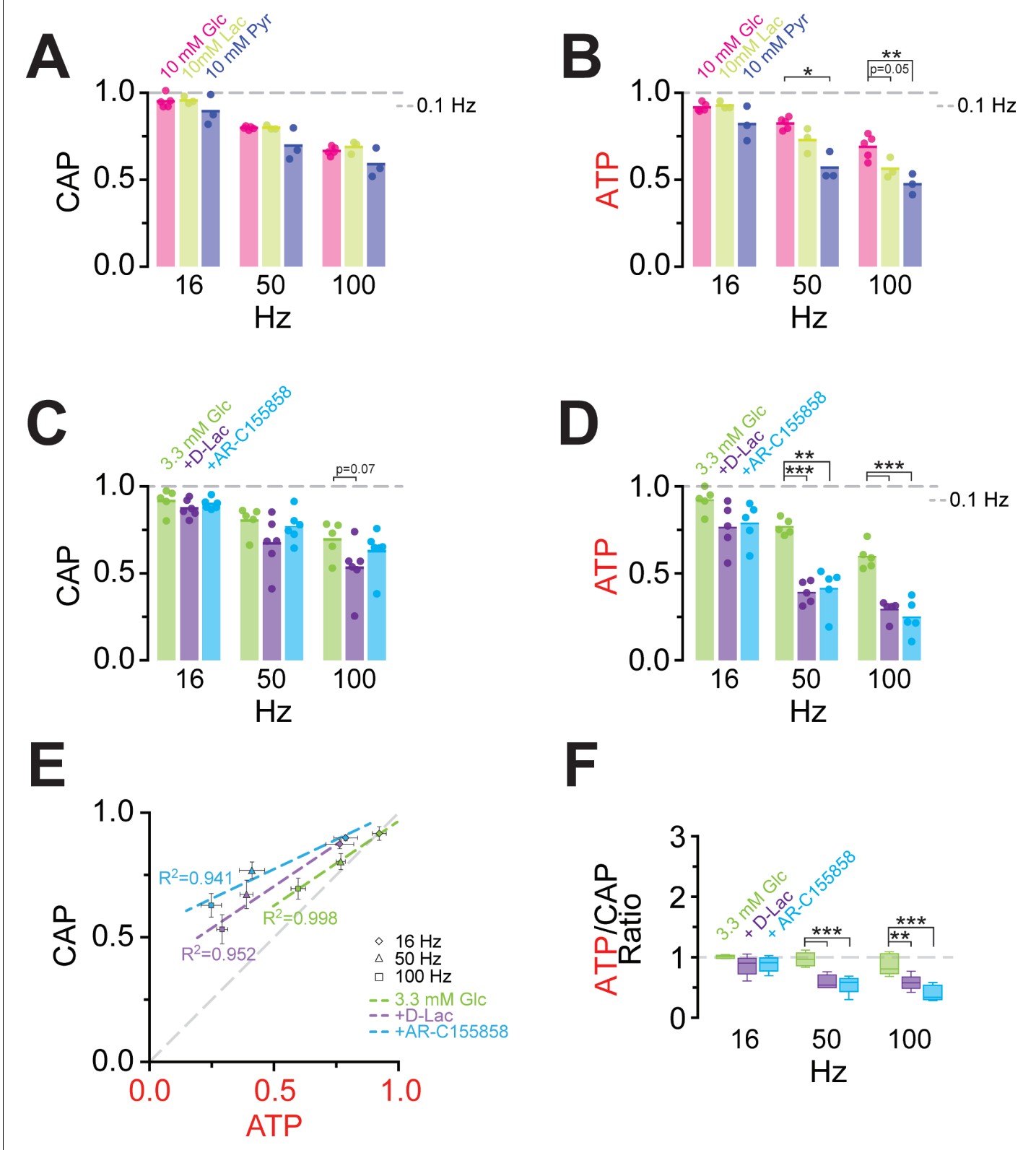

**Figure 5.** Energy metabolism of optic nerves depends on the type and concentration of substrates and involves lactate metabolism. (**A**) The comparison between CAP area decay of optic nerves incubated in 10 mM glucose aCSF (n = 5 nerves) versus optic nerves incubated either in 10 mM lactate or 10 mM pyruvate (n = 3 nerves) during HFS, shows no significant differences among the three substrates (p>0.05, Welch's t test). (**B**) In contrast, analysis of axonal ATP levels shows that at higher frequencies glucose is a better substrate to maintain axonal ATP levels. Same experiments

*Figure 5 continued on next page*

*Figure 5 continued*

as in panel **A**. (**C**) In the presence of glucose (3.3 mM; n = 5 nerves) as exogenous energy substrate, inhibition of lactate metabolism by D-lactate (20 mM; competitive inhibitor of endogenous L-lactate metabolism at MCTs and LDH, n = 6) or AR-C155858 (10 µM; MCT1 and MCT2 selective inhibitor, n = 6) does not significantly affect CAPs. (**D**) Analysis of ATP under the same conditions as in (**C**): ATP levels undergo a strong decrease at higher frequencies in the presence of D-lactate or AR-C155858. (n = 5 nerves for all conditions). The dashed lines in panels **A–D** at 1 show CAP size or ATP levels at 0.1 Hz stimulation frequency used for normalization. (**E**) Inhibition of metabolism of endogenously produced L-lactate in the presence of glucose as the sole exogenous energy substrate shifts the correlation of ATP and CAP to the upper left showing that ATP changes more strongly than CAP. (**F**) The ratio of ATP and CAP drop decreases significantly in the presence of inhibitors of lactate metabolism, confirming that ATP changes more strongly than CAP. Asterisks in (**A–D** and **F**) indicate significant differences among conditions: *p<0.05, **p<0.01, ***p<0.001; Welch's t test.

The following source data and figure supplements are available for figure 5:

**Source data 1.** Table containing data for *Figure 5*.

**Figure supplement 1.** Correlation of the amplitudes of CAP and ATP changes in the presence of lactate and pyruvate as exogenous energy substrates.

**Figure supplement 2.** Example of CAP traces before and after high-frequency stimulation (HFS) of optic nerves in the presence of inhibitors of lactate metabolism.

compartments prefer glucose or lactate as energy-rich substrates (*Pellerin and Magistretti, 2012*; *Dienel, 2012*; *Nave, 2010*; *Hirrlinger and Nave, 2014*). In the optic nerve, glucose and pyruvate/lactate can sustain CAP propagation. We found that axonal ATP levels were significantly lower during HFS when nerves were incubated with lactate or pyruvate as the sole energy substrate. This suggests that lactate/pyruvate alone does not support ATP production as well as glucose and confirms observations in cultured cerebellar granule cells with repetitive activation of NMDA-receptors (*Lange et al., 2015*).

Nevertheless, lactate metabolism contributes to axonal energy homeostasis, even in the presence of glucose, as evidenced by blocking lactate metabolism with D-lactate or AR-C155858. In the presence of these inhibitors, ATP decreased more strongly than CAP (*Figure 5E,F*), suggesting the presence of a 'safety window'. It also breaks the striking correlation of ATP and CAP observed under control conditions (*Figures 4E,F* and *5E,F*). One possible explanation is that changes in CAP are related to ATP consumption, while inhibition of lactate transport affects ATP production. Since the axonal stimulation protocol is unchanged, also the consumption of ATP should be similar in the presence of inhibitors of lactate metabolism. Therefore, the inhibition of lactate metabolism impairs ATP production and causes a lower steady-state level of ATP, but the consumption rates during HFS remain the same.

Furthermore, this finding suggests that 'endogenous' lactate (i.e. produced from glucose) is more efficient than lactate provided in the medium. One possibility is that glucose enters the nerve more rapidly than lactate, because glucose transporter expression is optimized in comparison to MCT1. Moreover, glucose and glycolysis intermediates are readily shuttled via gap junctional coupling between astrocytes and oligodendrocytes, with lactate being generated also close to the periaxonal space, whereas exogenous lactate has a more complex path of diffusion involving mostly extracellular space (*Hirrlinger and Nave, 2014*). We note that from an evolutionary point of view, such optimization of glucose utilization is of advantage for the brain, because the liver always aims at generating a constant blood glucose level, even under starvation conditions.

Taken together, combining ATP imaging and electrophysiology, we demonstrate that metabolic imaging is a very sensitive analysis tool for real-time monitoring of axonal energy homeostasis and the underlying neuron-glia interactions in electrically active fiber tracts. Applied to models of diseases this technology will be able to detect subtle perturbations of axonal energy homeostasis and allow addressing the hypothesis that energy deficits are an early and most likely causal event for neurodegeneration.

## Materials and methods

### Ethics statement

Animals were treated in accordance with the German Protection of Animals Act (TSchG §4 Abs. 3), with the guidelines for the welfare of experimental animals issued by the European Communities Council Directive 2010/63/EU as well as the regulation of the institutional 'Tierschutzkommission' and the local authorities (T04/13, T20/16; Landesdirektion Leipzig, LAVES Niedersachsen).

### Transgenic mice

Animals were bred in the animal facility of the Max-Planck-Institute for Experimental Medicine as well as the Medical Faculty of the University of Leipzig. Mice were housed in a 12 hr/12 hr light dark cycle with access to food and water ad libitum. The transgene construct was assembled in pTSC (*Hirrlinger et al., 2005*) containing *Thy1.2* promoter sequences. The plasmid pDR-GW AT1.03$^{YEMK}$ (*Bermejo et al., 2010*) containing the open reading frame of the ATP-sensor ATeam1.03$^{YEMK}$ (*Imamura et al., 2009*) was obtained from Wolf Frommer (via Addgene; plasmid #28004). The open reading frame of ATeam1.03$^{YEMK}$ was subcloned into the XhoI restriction site of pTSC using PCR. The linearized transgene was injected into fertilized mouse oocytes of the C57BL/6J mouse strain. Transgenic founders were identified using PCR-based genotyping on genomic DNA isolated from tail tips (primer 1: 5'-CGCTGAACTTGTGGCCGTTTACG-3'; primer 2: 5'-TCTGAGTGGCAAAGGACC TTAGG-3'). The mouse line has been registered at the RRID Portal (https://scicrunch.org/resources; RRID:MGI:5882597).

### Determination of copy number

Mouse genomic DNA was isolated from tail biopsies of four C57BL/6J and six ThyAT-mice with Invisorb Spin Tissue Mini Kit according to the manufacturer's instructions (Stratec Biomedical, Birkenfeld, Germany). Short FAM-labeled hydrolysis probes (UPL) were used for qPCR reactions (Roche Diagnostics GmbH, Mannheim, Germany). Primers and the UPL-probe were designed by ProbeFinder version 2.50 for Mouse (Roche Diagnostics; Primer Thy1s: TGCCGGTGTGTTGAGCTA; Thy1as: TGG TCCTGTGTTCATTGCTG; UPL 60; amplicon 73 bp). The genomic sequence of Nrg1 was used to calibrate for the amount of DNA (Primer Nrg1s: GGCTATAATGCTAACACAGTCCAA; Nrg1as: AG TGGATCGTAACAACACTGTCA; UPL 38; amplicon 61 bp) as described (*Besser et al., 2015*). 10 to 25 ng of genomic DNA was subjected to qPCR amplification to measure the amount of *Thy1.2* on a Light Cycler 480 system (Roche Diagnostics) according to the manufacturer's instructions. The copy number of the *Thy1.2*-ATeam1.03$^{YEMK}$ transgene was calculated using the $\Delta\Delta$Ct method compared to wild type mice carrying only the two endogenous alleles of the *Thy1* gene (*Besser et al., 2015*).

### Analysis of the sensor expression pattern

Adult mice were transcardially perfused with 4% formaldehyde solution (PFA, in phosphate buffered saline: 137 mM NaCl, 2.7 mM KCl, 8 mM Na$_2$HPO$_4$, 1.5 mM KH$_2$PO$_4$, pH 7.4) under deep anesthesia. The brain and the eyes were removed and post-fixed for 24 hr in the same fixative. 45 µm thick sections were cut on a vibratome (Leica VT1000 S, Nussloch, Germany) and slices were mounted directly after cutting with Vectashield embedding medium (Vectashield HardSet Mounting Medium, Vector Laboratories, Burlingame, CA, USA). From the eyes, retinal whole mounts were prepared.

For imaging of fixed brain slices and retina, confocal images were acquired on a LSM Olympus IX71 inverted microscope using an UPlanFL 10x/0.3 objective (Olympus, Hamburg, Germany) for overview images (*Figure 1A–C,G*) or an UApo/340 40x/1.35 oil objective (Olympus) for detailed images (*Figure 1D–F,H*), respectively. Microscopic images were acquired and processed using Olympus Software Fluoview v5.0. ATeam1.03$^{YEMK}$ sensor fluorescence was excited with a 488 nm argon laser and detected through a BA 510–540 nm emission filter (AHF Analysentechnik AG, Tübingen, Germany). For all images, a Kalman filter of two was used for denoising and images were acquired with 1024 × 1024 resolution (pixel size for overview images: 0.9 µm; pixel size for detail images: 0.35 µm). Z-stacks comprise 12–38 singles z-planes for overview images and 30–85 z-planes for detailed images, respectively, with distances between each z-plane of 2 µm (overview), 0.5 µm (*Figure 1D–F*) and 0.35 µm (*Figure 1H*). Single z-stacks were converted to maximum intensity projections (by using Fiji macro 'Flattening V2f.ijm'; generously provided by Jens Eilers; Jens-Karl.

Eilers@medizin.uni-leipzig.de) and for overview images maximum intensity projections of different positions (210 for *Figure 1A*; 154 for *Figure 1B*; 160 for *Figure 1C*; 49 for *Figure 1G*) were stitched by using Fiji software and a Fiji stitching plugin (*Preibisch et al., 2009*).

## Optic nerve preparation and electrophysiological recordings

For optic nerve experiments, mice were used in an age of 8 to 12 weeks. Optic nerves were excised from decapitated mice, placed into an interface perfusion chamber (Harvard Apparatus, Holliston, MA) and continuously superfused with artificial cerebrospinal fluid (aCSF). The perfusion chamber was continuously aerated by a humidified gas mixture of 95% $O_2$/5% $CO_2$ and experiments were performed at 37°C. Custom-made suction electrodes back-filled with aCSF were used for stimulation and recording as described (*Stys et al., 1991*; *Saab et al., 2016*). The stimulating electrode, connected to a battery (Stimulus Isolator 385; WPI, Berlin, Germany) delivered a supramaximal stimulus of 0.75 mA to the nerve evoking compound action potentials (CAP). The recording electrode was connected to an EPC9 amplifier (Heka Elektronik, Lambrecht/Pfalz, Germany). The signal was amplified 500 times, filtered at 30 kHz, and acquired at 20 kHz or 100 kHz. Before recording, optic nerves were equilibrated for at least 30 min in the chamber.

The CAP, elicited by the maximum stimulation of 0.75 mA, was recorded at baseline stimulation frequency at 0.1 Hz. During HFS a burst-stimulation was applied consisting of 100 stimuli at the given frequency (16, 50 or 100 Hz), separated by 460 ms, during which the CAP was recorded; HFS overall duration was 150 s, independent of the frequency.

## Imaging

Live imaging of optic nerves was performed using an up-right confocal laser scanning microscope (Zeiss LSM 510 META/NLO, Zeiss, Oberkochen, Germany) equipped with an Argon laser and a 63x objective (Zeiss 63x IR-Achroplan 0.9 W). The objective was immersed into the aCSF superfusing the optic nerve. Theoretical optical sections of 1.7 µm over a total scanned area of 66.7 µm x 66.7 µm (512 $\times$ 512 pixels) of the optic nerve were obtained every 10.4 s in three channels, referred as CFP (excitation 458 nm; emission 470–500 nm), FRET (Ex 458 nm; Em long pass 530 nm) and YFP (Ex 514 nm; Em long pass 530 nm).

## Solutions

Optic nerves were superfused by aCSF containing (in mM): 124 NaCl, 3 KCl, 2 $CaCl_2$, 2 $MgSO_4$, 1.25 $NaH_2PO_4$, and 23 $NaHCO_3$, which was continuously bubbled with carbogen (95% $O_2$/5% $CO_2$). This solution was substituted with the appropriate energy substrates as indicated for which the solution containing 10 mM glucose served as the standard solution in respect to pH and osmolarity. For glucose deprivation (GD), glucose was removed from the aCSF and substitute by sucrose (Merck Millipore, Darmstadt, Germany) to maintain the correct osmolarity. For mitochondrial blockage (MB), aCSF containing 10 mM glucose was supplemented with 5 mM sodium azide (Merck Millipore). For the combination of MB+GD glucose was substituted by sucrose and 5 mM sodium azide was added.

For HFS, aCSF was supplemented with different substrates. Glucose (Fluka BioChemika, Munich Germany) was used at three concentrations: 10 mM, 3.3 mM and 2 mM. L-lactate and pyruvate (Sigma-Adrich, Munich, Germany) were used at 10 mM. Inhibitors of lactate metabolism were added to a basal concentration of glucose of 3.3 mM: the MCT1 inhibitor AR-C155858 (Med Chem Express, Sollentuna, Sweden) was used at 10 µM; sodium D-lactate (Sigma-Aldrich) was used at 20 mM. All aCSF-based solutions were adjusted for the same pH, sodium concentration and osmolarity.

## Data analysis
### CAP

Optic nerve function was monitored quantitatively as the area under the supramaximal CAP. The CAP area is proportional to the total number of excited axons and represents a convenient and reliable means of monitoring optic nerve axon function (*Stys et al., 1991*; *Saab et al., 2016*). CAP was expressed as area under the CAP wave form in a range of approximately 1.5 ms, defined by the beginning of the response (typically at 0.2 ms after the stimulus) and approximately the end of the

second peak of the CAP wave form, thereby analyzing the first and second peak characteristic of the optic nerve evoked CAP. These two peaks are indicative for the large and medium-sized axons within the optic nerve, which are those that will mainly be resolved by confocal imaging of the ATP sensor expressed in axons. During high frequency action potential propagation, amplitudes of CAP peaks generally decrease, while peak latencies increase. The measured CAP area integrates these two functional changes of nerve conduction. It is, therefore, a parameter well suited to quantify over-all functional changes of the optic nerve. Data were normalized to the mean obtained during the initial 15 min (defined as baseline, i.e. no experimental condition applied). Results from several nerves were pooled, averaged, and plotted against time.

### ATP

The relative amount of ATP in the optic nerve was calculated in two steps. Initially, the mean intensity of FRET and CFP channel were extracted from the imaging time-series (in Fiji) and the ratio F/C was calculated. All fibers visible in the field of view were included within the analysis without any prior selection. The F/C ratio was then normalized to zero using the F/C value obtained by application of MB+GD at the end of each experiment, i.e. by depleting all axons from ATP. Furthermore, F/C values obtained during baseline recordings in 10 mM glucose at the beginning of each experiment were set to one. In all experiments, YFP fluorescence upon direct excitation of YFP was analyzed in parallel. Importantly, YFP fluorescence intensity remained stable throughout the experiments suggesting that the observed changes of the F/C ratio are most likely not due to artificial modulation e.g. by changes in intra-axonal pH.

### ATP and CAP parameters

For GD, MB and MB+GD conditions, $m$ coefficients for the slope-intercepts at the point of maximal change of the signal were used to calculate the CAP and ATP rate per second, during the initial decay and the recovery following reperfusion with aCSF containing 10 mM glucose. The time of the start of decay/recovery of the ATP and CAP curves were defined as the first time-point at which the slope-intercept of the signal was above a threshold value based on the standard deviation calculated at the baseline. For HFS analysis three parameters were considered for both CAP and ATP: (1) the overall amplitude obtained by averaging the data at the time points within the last 15 s of the stimulation (2) the rate of initial decay defined within the 45 s that followed the beginning of the stimulation (3) the rate of recovery defined within the 60 s that followed the end of the stimulation. Rates were calculated from the $m$ coefficients for the slope-intercepts in the defined time-range.

## Presentation of data

All data are presented either as mean ± s.e.m. or as stripe plots showing all data points and the mean (line within the plot). The number of optic nerves analyzed for each condition is given as n. As for no condition both optic nerves of one animal were used, the number of nerves is equal to the number of animals analyzed for each condition. If not indicated otherwise in the figure legends, data were statistically evaluated using Welch's t-test and assuming a normal distribution (*p<0.05; **p<0.01; ***p<0.001).

## Acknowledgements

Within the Max-Planck-Institute for Experimental Medicine, Göttingen, we thank the transgene core unit for performing oocyte injections, the mechanical workshop for construction of the imaging setup as well as the IT-department for help with handling huge datasets. We would like to thank Prof. Jens Eilers, Carl-Ludwig-Institute for Physiology, Leipzig, for providing custom written Fiji plugins and established work flows for image analysis.

## Additional information

### Competing interests

K-AN: Reviewing editor, *eLife*. The other authors declare that no competing interests exist.

## Funding

| Funder | Author |
|---|---|
| Deutsche Forschungsgemeinschaft | Johannes Hirrlinger |
| H2020 European Research Council | Klaus-Armin Nave |
| European Molecular Biology Organization | Aiman S Saab |

The funders had no role in study design, data collection and interpretation, or the decision to submit the work for publication.

## Author contributions

AT, Conceptualization, Formal analysis, Validation, Investigation, Visualization, Methodology, Writing—original draft, Writing—review and editing; ASS, Conceptualization, Formal analysis, Validation, Investigation, Methodology, Writing—review and editing; UW, WM, KK, Formal analysis, Investigation, Methodology, Writing—review and editing; GM, Investigation, Methodology, Writing—review and editing; HI, Resources, Writing—review and editing; K-AN, Conceptualization, Supervision, Writing—original draft, Project administration, Writing—review and editing; JH, Conceptualization, Data curation, Formal analysis, Supervision, Methodology, Writing—original draft, Project administration, Writing—review and editing

## Author ORCIDs

Wiebke Möbius, http://orcid.org/0000-0002-2902-7165
Klaus-Armin Nave, http://orcid.org/0000-0001-8724-9666
Johannes Hirrlinger, http://orcid.org/0000-0002-6327-0089

## Ethics

Animal experimentation: Animals were treated in accordance with the German Protection of Animals Act (TSchG §4 Abs. 3), with the guidelines for the welfare of experimental animals issued by the European Communities Council Directive 2010/63/EU as well as the regulation of the institutional "Tierschutzkommission" and the local authorities (T04/13, T20/16; Landesdirektion Leipzig, LAVES Niedersachsen).

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
