## [Decision Letter]

Thank you for submitting your article "Monitoring ATP dynamics in electrically active white matter tracts" for consideration by *eLife*. Your article has been reviewed by three peer reviewers, and the evaluation has been overseen by Gary Westbrook as the Senior Editor and Reviewing Editor. The following individuals involved in review of your submission have agreed to reveal their identity: Marc Freeman; Brian MacVicar; Peter Stys. The reviewers have discussed the reviews with one another and the Senior Editor has drafted this decision to help you prepare a revised submission.

Summary: All three reviewers thought that this was an important study and were enthusiastic about publication after appropriate revisions. The major issue was concerning the sample size on some of the experiments particularly the last set of data, and with some technical considerations raised by reviewers 2 and 3 that require your attention and potentially some additional data. This sample size issue is well-elaborated in the comments of reviewer 2 below. You may also want to consider whether the MCT experiments contribute sufficiently to the manuscript to warrant inclusion as described by reviewer 1. As the reviewer comments are rather straightforward and non-overlapping, the original text of their reviews is included below. Your revisions should modify the text of the revised manuscript so that readers of your paper that have similar concerns will be able to assess the data appropriately.

*Reviewer #1:*

This is a very important study that combines live imaging, a mouse model of disease, and metabolic experiments to explore some very basic and important questions in axonal biology. Nave and colleagues, for the first time to my knowledge, directly measure (simultaneously) compound action potentials (CAP) and axonal ATP levels in a dissected whole optic nerve preparation. They first generate what will be a very useful mouse for the field where an ATP sensor is expressed under control of the Thy1 promoter and identify lines that nicely label most neurons. They then track ATP levels and CAP after stimulation at different frequencies. In short, they provide a number of interesting observations that argue for a major role for ATP in axons in regulating the ability of neurons to fire. By performing glucose deprivation and blockade of mitochondrial function they explore the differences in how ATP levels fall, how that affects CAPs, and how quickly neurons recover the ability to fire CAPs. Through these experiments they answer many long standing questions and show: 1) axonal glycolysis is not enough to robustly sustain CAPs, mitochondrial function is key, 2) blocking mitochondrial function rapidly blocks CAPs, 3) glucose, pyruvate and lactate can all sustain CAP activity, and 4) blockade of MCT function or lactate transport only mildly affects CAP, but more strongly affects ATP. It is suprising how much ATP can be around, yet CAPs still fire. The final interesting, but not fully explained phenotype that is observed is the fact that PLP1 KO animals exhibit strong defects in ATP and CAPs, even in the presence of extracellular lactate or glucose. Why PLP1 KO animals are more sensitive than WT controls to high frequency stimulation is unclear. Finally, the MCT het KO experiments are not particularly informative. If easy, repeating the experiment in an oligo-specifc cKO situation would be very interesting. So, I'm not sure what that experiment really adds.

Overall the experiments are very well done, they establish an exciting new system to examine axonal ATP levels and CAPs in an intact nerve preparation, and they provide some of the most satisfying experiments I know of that address the sufficiency of glucose, lactate, and pyruvate for sustaining axonal ATP and CAPs.

*Reviewer #2:*

This study follows from an impressive series of discoveries over the past few years from this research team on the roles for oligodendrocytes in providing energy substrates to maintain axonal functions and survival. They have created a very impressive and clever animal model of disease with a novel strategy to detect ATP levels in neurons using a FRET sensor for ATP. This potentially provides the authors with a method to examine the changes in ATP levels and metabolically driven alterations in ATP at different times leading up to clear manifestations of the disease phenotype. The authors observed remarkable parallels between CAP and ATP that are very reassuring for the interpretation of their altered relationship in the disease model. The authors have also been able to test the impact of spiking frequencies on ATP levels as a parallel to several previous studies that have shown decreasing ATP levels reduces action potential firing in optic nerves. Again, the authors found striking correlations between the spike frequency, the decrease in ATP levels and the impact on compound action potentials as shown in Figure 4.

1) The authors show in Figure 3 that there is a striking correspondence between decreases in both CAP and ATP during high frequency stimulation. However, the relationship appears to break down when changes are compared in fairly high concentrations of D-lactate or in the MCT_1/2_ inhibitor AR-C155858. Why is this?

2) The pH dependence from the original Imamura PNAS paper is complex and varies depending on the ATP concentration. The excitation for FRET at 458 is a longer wavelength than the Imamura paper (435 nm). At 435 nm there is selective excitation of CFP but at 458 there is some excitation of YFP. Therefore, at this wavelength there is a minor component of YFP direct excitation contributing to the FRET signal. This is a potential worry that the authors could eliminate by testing the pH dependence using their system. The authors mention normalizing some of the results with changes in YFP fluorescence changes. However, this is not clearly explained in the Methods. Was this always done? Could changes in pH play a role in altering the FRET signals in the animals’ disease models at different times of development. I suspect not but the authors should include some discussion on these points.

3) The activity dependent changes in ATP signals and therefore ATP changes during high frequency spiking are very interesting. However, the results in Figure 6 are the critical data for concluding there are early changes in the disease model. There are small changes in the disease model in the ATP FRET signal at 50 Hz stim in glucose and in the lactate groups. However, these results are only from 3 nerves in each group (Figure 6D and F). This small number seems too low particularly as there is one quite divergent value in the 50 Hz lactate category. In addition the authors should include both the number of animals and the number of nerves (e.g. was more than one nerve used from one animal). The text in the Results section states there were 4 nerves in each set but there are only 3 points in the bar charts and the figure legends says 3 nerves.

4) Subsection “PLP loss in oligodendrocytes impairs axonal ATP homeostasis”, last paragraph: The authors hint there is a non-significant trend towards reduced ATP levels in the optic nerves from *Slc16a1* gets but I don't see this and it is not significant. Therefore, I would conclude that there is no effect if it is not statistically significant. In the conclusions "Applied to a mouse model of X-linked spastic paraplegia or SPG-2 we were able to detect an effect of mutant oligodendrocytes on axonal energy metabolism and physiology several months before these animals develop widespread axonal loss and spastic paraplegia. Compared at a preclinical stage to wildtype nerves, ATP homeostasis is impaired in axons conducting at high frequency, but appears normal under resting conditions." The data has the minimum n values to state that ATP homeostasis is impaired. The authors should increase the number of experiments to provide more robust evidence. There may be variations in the handling etc. that may distort a few of the values in such a few experiments.

*Reviewer #3:*

This is a very interesting and elegant study of CNS axon energy metabolism with a focus on axonal ATP using using a novel axonally-expressed FRET-based fluorescent ATP reporter. Importantly, they shed important light on how genetic defects of myelin production paradoxically result in axonal degeneration.

1) "while after 2min of MB+GD the F/C-ratio was reduced by 35.1% ± 1.7% (n=19 121 nerves; Figure 2)": p values and significance need to be cited.

2) Subsection “Data analysis” – CAP analysis: I agree with authors' choice of analyzing CAP area up to 1.5ms, which should include large and medium-sized axons, the ones that will be resolvable by confocal imaging. Authors should explain this point in Methods.

3) Figure 3: authors should include a supplementary figure showing representative CAP waveforms and how they decay with the various energy deprivation paradigms, which may differ.

4) Given points 2 and 3 above, raises a potential difficulty: consider Figure 4 for example, where CAP changes also include a significant lengthening of the waveform: by restricting the area analysis to 1.5ms, while still imaging ATP FRET signals in medium-sized axons by confocal, there may arise an artifactual disconnect between CAP area and axonal ATP levels, merely because CAP lengthening will move significant signal beyond their 1.5ms integration window. Authors should explain how they dealt with this potential limitation and how it may have affected their results.

5) Subsection “ATP homeostasis of myelinated axons depends on lactate metabolism”, last paragraph: "…plus 20mM D-lactate.…": authors should clarify that physiological lactate is the L isomer for readers who may not know.

6) Discussion, third paragraph: given their ATP FRET reporter, can the authors give a rough estimate of absolute ATP concentrations in axons? This would be an interesting piece of reference data for the field.

---

## [Author Response]

*Reviewer #1:*

*This is a very important study that combines live imaging, a mouse model of disease, and metabolic experiments to explore some very basic and important questions in axonal biology. Nave and colleagues, for the first time to my knowledge, directly measure (simultaneously) compound action potentials (CAP) and axonal ATP levels in a dissected whole optic nerve preparation. They first generate what will be a very useful mouse for the field where an ATP sensor is expressed under control of the Thy1 promoter and identify lines that nicely label most neurons. They then track ATP levels and CAP after stimulation at different frequencies. In short, they provide a number of interesting observations that argue for a major role for ATP in axons in regulating the ability of neurons to fire. By performing glucose deprivation and blockade of mitochondrial function they explore the differences in how ATP levels fall, how that affects CAPs, and how quickly neurons recover the ability to fire CAPs. Through these experiments they answer many long standing questions and show: 1) axonal glycolysis is not enough to robustly sustain CAPs, mitochondrial function is key, 2) blocking mitochondrial function rapidly blocks CAPs, 3) glucose, pyruvate and lactate can all sustain CAP activity, and 4) blockade of MCT function or lactate transport only mildly affects CAP, but more strongly affects ATP. It is suprising how much ATP can be around, yet CAPs still fire.*

*The final interesting, but not fully explained phenotype that is observed is the fact that PLP1 KO animals exhibit strong defects in ATP and CAPs, even in the presence of extracellular lactate or glucose. Why PLP1 KO animals are more sensitive than WT controls to high frequency stimulation is unclear. Finally, the MCT het KO experiments are not particularly informative. If easy, repeating the experiment in an oligo-specifc cKO situation would be very interesting. So, I'm not sure what that experiment really adds.*

We very much agree with these points and have changed the manuscript accordingly (see also the response to referee 2 below).

The original "motivation" to include Plp1 KO and MCT1 het data at this stage was twofold. We had thought this could provide a missing "proof-of-principle" that the recently discovered function of oligodendrocytes in metabolic support of axon function (Fünfschilling et al., Nature 2012; Lee et al., Nature 20012) and our much older observation that oligodendrocytes are required for axonal integrity independent of myelin itself (Griffiths et al., Science 1998; Lappe-Siefke et al., Nat Genet. 2003) are two sides of the same coin. However, we realize that it is better to make this point with a full-sized data set once we have generated the necessary mouse numbers. Additional mouse mutants will be available only in about 6 months from now.

Similarly, the motivation to include MCT1 heterozygous mice here was the attempt to bridge to the Nature paper by Lee et al. (2012), assuming axonal degeneration in the optic nerve is preceded by an early metabolic phenotype. Surprisingly, no differences in axonal ATP levels were detected yet between MCT heterozygous mice and controls (see original Figure 6—figure supplement 1). This is likely no contradiction to Lee et al. as we investigated optic nerves at much earlier time points. Most likely, we will make different observations later in life. We also point out that the most degeneration-prone axons of Lee et al. 2012 (and also in Griffiths et al., 1998) are those of small radial caliber, whereas we selected on average larger axons for FRET analysis and CAP recordings.

We have now followed the advice of referee 1 and have removed Figure 6—figure supplement 1 from the final manuscript. The suggestion to include oligodendrocyte-specific MCT1 cKO mice awaits the collaboration with our US colleagues who may be willing to provide us with MCT1-floxed mice.

*Reviewer #2:*

*[…] 1) The authors show in Figure 3 that there is a striking correspondence between decreases in both CAP and ATP during high frequency stimulation. However, the relationship appears to break down when changes are compared in fairly high concentrations of D-lactate or in the MCT_1/2_ inhibitor AR-C155858. Why is this?*

We presume this point refers to Figure 4 and Figure 5 (notably Figure 5). Indeed, this is an intriguing finding, which confirms CAP area on one hand as a proxy for axonal ATP levels; however, under metabolic stress (high frequency stimulation in combination with inhibition of lactate metabolism) CAP recordings appear less vulnerable than ATP measurements.

The concentration of ATP in the axon is affected by both the usage of ATP and its production. Changes in CAP might be more related to ATP consumption, while inhibition of lactate transport and metabolism most likely mainly affects ATP production. As the stimulation protocol is identical in controls and after application of D-lactate or AR-C155858, we assume that consumption of ATP is similar in these conditions. Therefore, we think the most likely explanation is that inhibition of lactate metabolism impairs ATP production, resulting in a lower level of ATP as observed in our experiments.

Formally, we cannot exclude other mechanisms, such as the increasing levels of extracellular K^+^ or altered Ca^2+^ signaling. However, all changes in CAP are likely related to ATP consumption, while inhibition of lactate transport would affect ATP production. We also note that the CAP waveform is not affected by the presence of these inhibitors (see new Figure 5—figure supplement 2), suggesting that axonal conduction properties not dramatically impaired.

A discussion of this issue has been added to the last paragraph of the subsection “ATP homeostasis of myelinated axons depends on lactate metabolism”.

*2) The pH dependence from the original Imamura PNAS paper is complex and varies depending on the ATP concentration. The excitation for FRET at 458 is a longer wavelength than the Imamura paper (435 nm). At 435 nm there is selective excitation of CFP but at 458 there is some excitation of YFP. Therefore, at this wavelength there is a minor component of YFP direct excitation contributing to the FRET signal. This is a potential worry that the authors could eliminate by testing the pH dependence using their system. The authors mention normalizing some of the results with changes in YFP fluorescence changes. However, this is not clearly explained in the Methods. Was this always done? Could changes in pH play a role in altering the FRET signals in the animals’ disease models at different times of development. I suspect not but the authors should include some discussion on these points.*

We agree this is an important point that indeed might affect experiments using FRET based sensors for metabolites.

In our experiments, we had always determined the ATP concentration by a ratiometric measurement, i.e. FRET / CFP = (excitation of CFP / emission of YFP) divided by (excitation of CFP / emission of CFP). YFP-fluorescence is considered the most pH sensitive part of the sensor and YFP fluorescence should change in case of significant pH changes. We had therefore also monitored (routinely) the YFP fluorescence following direct YFP excitation at 514 nm to reveal such pH effects, which we did not detect. In fact, the YFP signal was very stable in those experiments, which is shown for the experiments with application of mitochondrial blockage plus glucose deprivation in Figure 1 of the original manuscript. Thus, it is unlikely that the ATP-signal is mainly a modulation of the readout due to pH changes. We have added a clarifying statement in the Methods section (subsection “ATP”).

In response to this point, we have added more experiments analyzing YFP fluorescence during glucose deprivation, mitochondrial blockage, and during high frequency stimulation (see new Figure 3—figure supplement 2, Figure 4—figure supplement 5) and added a corresponding description in the Results sections “Kinetics of ATP decline during energy deprivation” and “ATP homeostasis of myelinated axons depends on lactate metabolism”. This shows that also under more challenging conditions, pH changes did not significantly affect our ATP recordings.

*3) The activity dependent changes in ATP signals and therefore ATP changes during high frequency spiking are very interesting. However, the results in Figure 6 are the critical data for concluding there are early changes in the disease model. There are small changes in the disease model in the ATP FRET signal at 50 Hz stim in glucose and in the lactate groups. However, these results are only from 3 nerves in each group (Figure 6D and F). This small number seems too low particularly as there is one quite divergent value in the 50 Hz lactate category.*

We completely agree and have removed this data set as too preliminary from the revised manuscript. See also our response to reviewer 1 (above).

*In addition the authors should include both the number of animals and the number of nerves (e.g. was more than one nerve used from one animal).*

In all experiments, we have used only one nerve from each animal for each condition of experiment shown, i.e. the number of nerves equals to the number of animals. We have added a clarifying sentence in the Methods section “ATP”.

*The text in the Results section states there were 4 nerves in each set but there are only 3 points in the bar charts and the figure legends says 3 nerves.*

We assume that this refers to Figure 5, showing D-Lac and AR-C155858 data, where CAP data were given for 4 nerves while ATP levels are shown only for 3 nerves. Here, ATP imaging of one nerve could not be properly analyzed (due to loosing of focal plane) while CAP recordings were normally analyzed. We have now repeated these experiments and n numbers are now for both inhibitors increased to n=6 for CAP and n=5 for ATP. We have clarified this point in the last paragraph of the Results section “ATP homeostasis of myelinated axons depends on lactate metabolism” and figure legend of Figure 5.

*4) Subsection “PLP loss in oligodendrocytes impairs axonal ATP homeostasis”, last paragraph: The authors hint there is a non-significant trend towards reduced ATP levels in the optic nerves from Slc16a1 gets but I don't see this and it is not significant. Therefore, I would conclude that there is no effect if it is not statistically significant.*

As suggested by reviewer 1, the too preliminary data from mutant mice (original Figure 6) have been removed from the revised manuscript. See also above.

*In the conclusions "Applied to a mouse model of X-linked spastic paraplegia or SPG-2 we were able to detect an effect of mutant oligodendrocytes on axonal energy metabolism and physiology several months before these animals develop widespread axonal loss and spastic paraplegia. Compared at a preclinical stage to wildtype nerves, ATP homeostasis is impaired in axons conducting at high frequency, but appears normal under resting conditions." The data has the minimum n values to state that ATP homeostasis is impaired. The authors should increase the number of experiments to provide more robust evidence. There may be variations in the handling etc. that may distort a few of the values in such a few experiments.*

As suggested by reviewer 1, the too preliminary data from mutant mice (original Figure 6) have been removed from the revised manuscript. See also above.

However, we have performed additional experiments to increase the n of wildtype mice for experiments studying the effect of D-Lac and AR-C155858 (see revised Figure 5; now n=6 nerves for CAP and n=5 nerves for ATP) and the results confirmed our previous data.

*Reviewer #3:*

*This is a very interesting and elegant study of CNS axon energy metabolism with a focus on axonal ATP using using a novel axonally-expressed FRET-based fluorescent ATP reporter. Importantly, they shed important light on how genetic defects of myelin production paradoxically result in axonal degeneration.*

We thank the reviewer for his positive comments.

*1) "while after 2min of MB+GD the F/C-ratio was reduced by 35.1% ± 1.7% (n=19 121 nerves; Figure 2)": p values and significance need to be cited.*

Respective p values and significance values have been added to the text and the figure.

*2) Subsection “Data analysis” – CAP analysis: I agree with authors' choice of analyzing CAP area up to 1.5ms, which should include large and medium-sized axons, the ones that will be resolvable by confocal imaging. Authors should explain this point in Methods.*

As requested, we have added an explanatory statement in the Methods section “CAP”. We also now indicate these time windows in Figure 3—figure supplement 1 and Figure 4—figure supplement 1.

*3) Figure 3: authors should include a supplementary figure showing representative CAP waveforms and how they decay with the various energy deprivation paradigms, which may differ.*

As requested, a new supplementary figure panel was added to the revised manuscript, which shows the CAP waveforms for all three paradigms of energy deprivation (glucose deprivation, mitochondrial blockage, and both) for different time points. See new Figure 3—figure supplement 1.

*4) Given points 2 and 3 above, raises a potential difficulty: consider Figure 4 for example, where CAP changes also include a significant lengthening of the waveform: by restricting the area analysis to 1.5ms, while still imaging ATP FRET signals in medium-sized axons by confocal, there may arise an artifactual disconnect between CAP area and axonal ATP levels, merely because CAP lengthening will move significant signal beyond their 1.5ms integration window. Authors should explain how they dealt with this potential limitation and how it may have affected their results.*

We thank the reviewer for raising this important issue. How to quantify "CAP areas" is indeed a complicated issue and all paradigms have their own pros and cons. When using a "fixed" time window (as done in our present analysis) there is no need to define the borders between different peaks, which can be difficult and arbitrary, or to define the time point when the CAP area reaches baseline level again (note the asymptotic curve). However, as pointed out this approach is risking an overestimation of the loss of CAP area with increasing peak latencies (which is indeed the case for the high frequency stimulation depicted in Figure 4). Thus, the CAP areas measured with a "fixed" time window integrate two functional changes of nerve conduction: the decrease of peak amplitudes and the increase in peak latency. It is, therefore, a parameter well suited to quantify overall functional changes of the optic nerve. As already discussed in our original manuscript, ATP imaging was performed in a single optical plane comprising typically around 30 axons with a random proportion of large and medium sized axons, while CAP recordings reflect the summation of action potential propagation of virtually all axons within the optic nerve. We therefore never did attempt to relate changes in single CAP peaks to ATP changes. However, the results obtained are consistent between different nerves irrespectively of the selected imaging region (and therefore the proportion of large and medium-sized axons), suggesting that both parameters used are a good approximation for the functional properties of the optic nerve.

We have a clarifying sentence Methods section “CAP”.

*5) Subsection “ATP homeostasis of myelinated axons depends on lactate metabolism”, last paragraph: "…plus 20mM D-lactate.…": authors should clarify that physiological lactate is the L isomer for readers who may not know.*

We agree and have added an explanatory note on D-lactate and L-lactate in the third paragraph of the subsection “ATP homeostasis of myelinated axons depends on lactate metabolism”.

*6) Discussion, third paragraph: given their ATP FRET reporter, can the authors give a rough estimate of absolute ATP concentrations in axons? This would be an interesting piece of reference data for the field.*

We fully agree with the reviewer that the absolute concentration of ATP in axons is a missing information. Indeed, we have spent significant effort in trying to calibrate the ATP sensor, which, however, is more difficult than expected:

i) Calibration should be performed in the same tissue as the measurement itself (optic nerve axons) in order to avoid errors caused by sample specific optical properties i.e. scattering light in anisotropic tissue.

ii) For calibration, ATP concentrations need to be controlled from outside, e.g. by permeabilizing the membrane, but without leakage of the sensor itself. So far we were unsuccessful to do this, because myelin membranes require quite harsh permeabilization conditions.

iii) Even if permeabilization is successful, it is difficult to achieve an equal distribution of ATP inside and outside a cellular compartment. We note that ATP is a negatively charged molecule that does not diffuse freely into axons through a membrane with a negative membrane potential.

iv) Numerous intracellular enzymes use ATP and it is almost impossible to block all ATP consuming processes (the main consumer Na^+^/K^+^-ATPase can be blocked, but many others not). The risk remains that ATP inside the axon is lower than outside. For all these reasons we have not included absolute ATP concentrations in our original manuscript.

However, as we readily see changes of the ATP signal in the course of an experiment, it is very likely that the basal ATP level is within the dynamic range of the sensor (Kd = 1.2 mM, Imamura et al., 2009), possibly close to 2 mM. This concentration would also be in line with the recently reported concentration of free ATP in cultured neurons (1.4 mM; Rangaraju et al., 2014) and synaptic boutons of cultured neurons (2-4mM; Pathak et al., 2015).

We have added a sentence explaining this issue in the second paragraph of the Discussion.

To really solve this issue in the not too distant future, we have started analyzing the ATP sensor signals by fluorescent life time imaging (FLIM). This approach should allow a more precise calibration and absolute measurement of ATP concentrations in optic nerves, as the FLIM readouts are expected to not be that affected by the cellular environment.